# Study of the Deposition Formation Mechanism in the Heat Exchanger System of RHF

**Yuzhu Pan, Xuefeng She \*, Jingsong Wang and Yingli Liu** 

State Key Laboratory of Advanced Metallurgy, University of Science and Technology Beijing (USTB),
30 Xueyuan Road, Haidian District, Beijing 100083, China; panyuzhuustb@163.com (Y.P.);
wangjingsong@ustb.edu.cn (J.W.); liuyingli112@126.com (Y.L.)
\* Correspondence: shexuefeng@ustb.edu.cn; Tel.: +86-138-1156-9346

**Abstract:** In rotary hearth furnace (RHF) production, the heat transfer system will produce deposition, which blocks the exhaust channel. The formation of deposition will affect RHF production. In this study, the formation mechanism of deposition was determined through chemical composition analysis, XRD and SEM-EDS: the main cohered phase in the deposition was KCl and the secondary cohered phase was $ZnFe_2O_4$; the $ZnFe_2O_4$ had become solid since it was formed with a porous structure and it cohered other substances; the $ZnFe_2O_4$ exhibited stronger cohering strength than KCl, due to a different cohering mechanism. In contrast, the KCl played a significant role in the deposition on the heat exchanger wall. A new process was proposed to avoid the deposition formation. This process could eliminate the deposition in the heat transfer system of RHF and improve the utilization of metallurgical waste.

**Keywords:** RHF; deposition; KCl; $ZnFe_2O_4$; cohering mechanism

## 1. Introduction

Large amounts of waste are generated in ironmaking and steelmaking processes, such as sinter dusts, blast furnace dusts and converter dusts. These wastes are valuable because they contain Fe and C. Adversely, these dusts are not directly used in blast furnace production because they contain high levels of Pb, Zn, K and Na, which are harmful to blast furnace production. The direct reduction of the ironmaking process of rotary hearth furnace (RHF) is an effective method for the treatment of metallurgical wastes and it has received significant attention [1–4]. Five RHFs for the treatment of metallurgical wastes have existed in Kimtsu, Hikari and Hirohata in Japan since 2000 [5,6]. In China, the Laiwu Steel, the Maanshan Iron and Steel, the Shangan, the Rizhao Iron and Steel and other enterprises have built RHFs for dealing with metallurgical wastes. Usually, the wastes are turned into carbon-bearing pellets, from which C is effectively utilized for the reduction of metal elements. Pb and Zn through the reduction and re-oxidation into the gas phase form the secondary dust. KCl will volatilize into the gas phase and Fe will form direct reduced iron (DRI). Zn, Pb and KCl are obtained through the secondary dust treatment. The DRI can be used for Blast furnace process as confirmed by She [7]. Through the RHF process, the separation of harmful elements and Fe is achieved, in order for the metallurgical wastes to be effectively utilized.

In order to improve the RHF energy efficiency, the gas waste heat was recovered. In the waste heat recovery process, Zn, Pb, K and Fe produce deposition, which blocks the exhaust channel. The production of RHF is affected, because the occurrence of deposition reduces the heat transfer efficiency and increases the furnace pressure. In the past, regarding the RHF process study, researchers were often concerned about the metal element reduction and energy efficiency [8–12], while the effect of deposition on the heat transfer system for RHF was ignored. In this study, the deposition was

analyzed through chemical analysis, XRD, as well as SEM-EDS analysis and experiments. Consequently, the formation mechanism of deposition was obtained.

## 2. Experimental Procedure

The deposition sample was obtained from the RHF waste heat recovery system, as presented in Figure 1. In this RHF, sintering dust, pellet dust and blast furnace dust were utilized as raw materials to produce metalized pellets. The RHF production cycle was 15 min. The RHF production is presented in Figure 2. The waste heat recovery system consisted of a high temperature heat exchanger and a low temperature heat exchanger. The inlet temperature of the high temperature heat exchanger was approximately 1200 °C, the outlet temperature was approximately 700 °C, while the flue gas flow rate was 5–6 × 10³ m³/h. The deposition mainly occurred at the wall end of the high temperature heat exchanger.

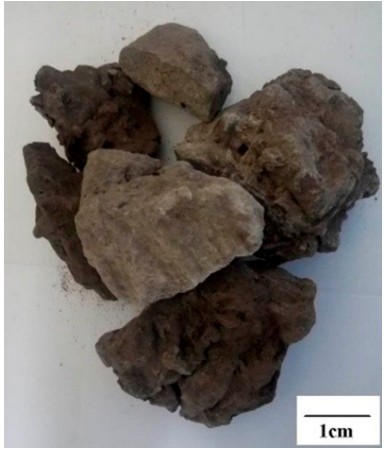

**Figure 1.** Deposition samples of RHF waste heat recovery system.

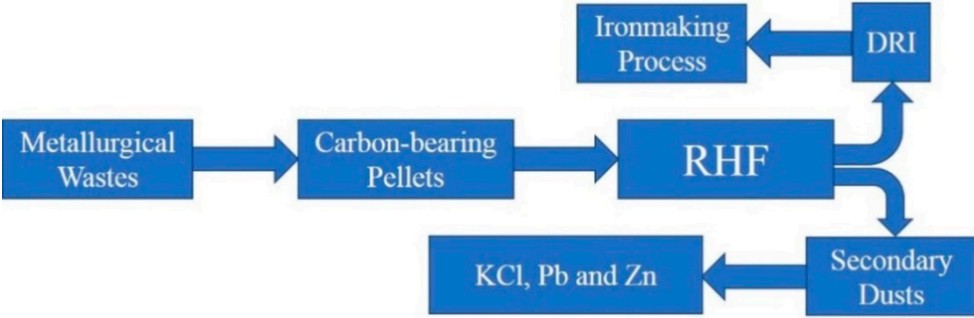

**Figure 2.** Waste treatment of RHF process.

In order to determine the deposition formation mechanism, the deposition samples were analyzed through composition analysis, XRD and SEM-EDS, while the determination of cohered phase experiments were carried out according to the composition and XRD analyses. The results of composition analysis and XRD are presented in Table 1 and Figure 3, respectively. The conducted experiments are presented in Table 2. The main elements in the deposition are Zn, Pb, K and Fe. During the design experiment, Zn, Pb, Fe and K are expressed in the form of ZnO, PbO, $Fe_2O_3$ and KCl according to the actual elements' ratio of deposition. However, some of the Zn elements exist in the form of $ZnCl_2$ and other minor components such as CaO, $SiO_2$ and C are present, which may slightly affect the composition reported in Table 2. The raw materials were pure chemical reagents. The weight of each raw material was 20 g and the raw materials were thoroughly mixed. Mixed raw materials were placed in a corundum crucible and heated in a muffle furnace for 15 min. The experimental temperature was 700 °C. The purpose of experiments 1–4 was to determine the cohered phases, whereas their

composition was designed according to the compositions of Table 1. The samples 1–4 were tested with XRD. According to Figure 3, the deposition structure consisted of five simple compounds and a complex compound. Therefore, it was necessary to study the effect of $ZnFe_2O_4$ for the deposition formation. The experiment 5, in order to study the effect of $ZnFe_2O_4$ on the deposition formation, was carried out twice: one of the samples was subjected to water quenching, while the other sample was cooled at room temperature. The two samples were analyzed through XRD and SEM-EDS. A high gas flow rate existed in heat exchanger system of RHF. The deposition product must have the ability to resist producing mechanical damage to the wall stability of the heat exchanger. Therefore, a drop strength test was designed to detect the bond strength of various substances. Subsequently to the sample obtained in experiment 8–13 room temperature cooling, the sample was allowed to fall into a sufficiently large corundum crucible at a height of 20 cm. Following the sample drop, the powder was collected and the bond strength of the sample was $N$ times/20 cm, subsequently to falling $N$ times when the powder exceeded 50% of the total mass.

**Table 1.** Chemical composition of deposition (mass %).

| Chemical Compositions % | CaO | SiO$_2$ | Zn | Pb | K | C | TFe |
|---|---|---|---|---|---|---|---|
| Deposition | 1.62 | 1.08 | 36.5 | 8.4 | 11.57 | 0.66 | 12.84 |

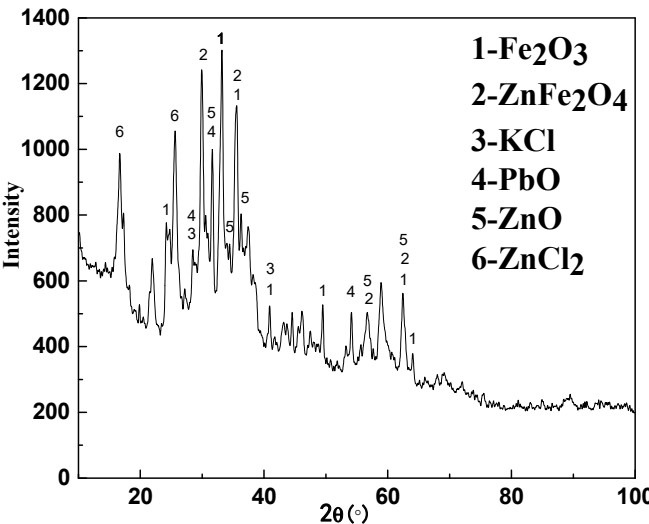

**Figure 3.** XRD pattern of deposition products.

**Table 2.** Chemical composition of experimental materials (mass %).

| Experiments | ZnO | PbO | Fe$_2$O$_3$ | KCl |
|---|---|---|---|---|
| 1 | 84.38 | 15.62 | - | - |
| 2 | 62.89 | 11.63 | 25.48 | - |
| 3 | 63.94 | 11.83 | - | 24.22 |
| 4 | 50.81 | 9.4 | 20.53 | 19.26 |
| 5 | 50.31 | - | 49.46 | - |
| 6 | 80.78 | 14.22 | - | 5 |
| 7 | 76.43 | 13.57 | - | 10 |
| 8 | 72.29 | 12.71 | - | 15 |
| 9 | 67.86 | 12.14 | - | 20 |
| 10 | 80.78 | 14.22 | 5 | - |
| 11 | 76.43 | 13.57 | 10 | - |
| 12 | 72.29 | 12.71 | 15 | - |
| 13 | 67.86 | 12.14 | 20 | - |

## 3. Results and Discussion

### 3.1. Analysis Results of Deposition

The chemical composition of the deposition and the XRD results are given in Table 1 and Figure 3. According to Table 1 and Figure 3, it could be observed that the deposition product was mainly composed of ZnO, PbO, KCl, $Fe_2O_3$, $ZnCl_2$ and $ZnFe_2O_4$. The ZnO and PbO were derived from the reduction and re-oxidation of Zn and Pb in the carbon-bearing pellets. The KCl was volatilized directly from the carbon-bearing pellets in the high temperature furnace of the RHF. The $ZnCl_2$ was not present in the carbon-bearing pellets, while its formation was more complicated. The Cl in $ZnCl_2$ was derived from KCl. The key to the formation of $ZnCl_2$ was the decomposition of KCl. Certain studies demonstrated that the corrosion of pure iron in oxidizing atmosphere with KCl vapor could result in the following reaction [13–15]:

$$2Fe_2O_3 + 2KCl_{(g)} + \frac{1}{2}O_2 = K_2Fe_4O_7 + Cl_{2(g)} \tag{1}$$

A high amount of $K_2Fe_4O_7$ was detected on the etched iron surface, indicating that the Reaction (1) could separate the K and Cl. Unfortunately, the thermodynamic data of Reaction (1) were not found. The vapor of Zn and $Cl_2$ can react as:

$$Zn_{(g)} + Cl_{2(g)} = ZnCl_{2(g)} \tag{2}$$

The reason for the $ZnCl_2$ formation in the deposition product could be explained by Reactions (1) and (2). The $Fe_2O_3$ was derived from the carbon-bearing pellets. The carbon-bearing pellets would produce cracks or even break to produce high-sized particles of iron-containing dust in the heating and reduction process. Blast furnace dust, which contained a certain amount of $ZnFe_2O_4$, was one of the raw materials for RHF. Yet, the $ZnFe_2O_4$ would be destroyed during the reduction of carbon pellets as presented through Reactions (3)–(5):

$$ZnFe_2O_4 + C = ZnO + 2FeO + CO_{(g)} \tag{3}$$

$$ZnFe_2O_4 + CO_{(g)} = ZnO + 2FeO + CO_{2(g)} \tag{4}$$

$$ZnFe_2O_4 + 2Fe = 4FeO + Zn_{(g)} \tag{5}$$

It could be determined that the $ZnFe_2O_4$ was produced by the contact between $Fe_2O_3$ and ZnO during deposition. The formation mechanism of $ZnFe_2O_4$ and its role in the deposition was described in Sections 3.2 and 3.3.

The typical morphology of the deposition is presented in Figure 4. Apparently flat, porous and transitional areas on the deposition surface existed, with certain bright white areas being distributed on the deposition product surface. In Figure 4, the left side was near the end of the heat exchanger wall and the right side was the end near the center of the heat exchanger.

The EDS maps of the transitional and flat areas are presented in Figure 5. In Figure 5, the sample was taken from the heat exchanger wall surface and the upper part of the sample was attached to the heat exchanger wall. It could be observed that the distribution areas of Cl (Figure 5b) and K (Figure 5c) overlapped, combining the chemical composition of Table 1 and the results of Figure 3. The XRD result demonstrated that the cubic particles of the transitional region were KCl, while the other locations of the transitional areas were filled with ZnO. The flat areas were mainly composed of ZnO particles, while the bright white areas were mainly composed of PbO (Figure 5f).

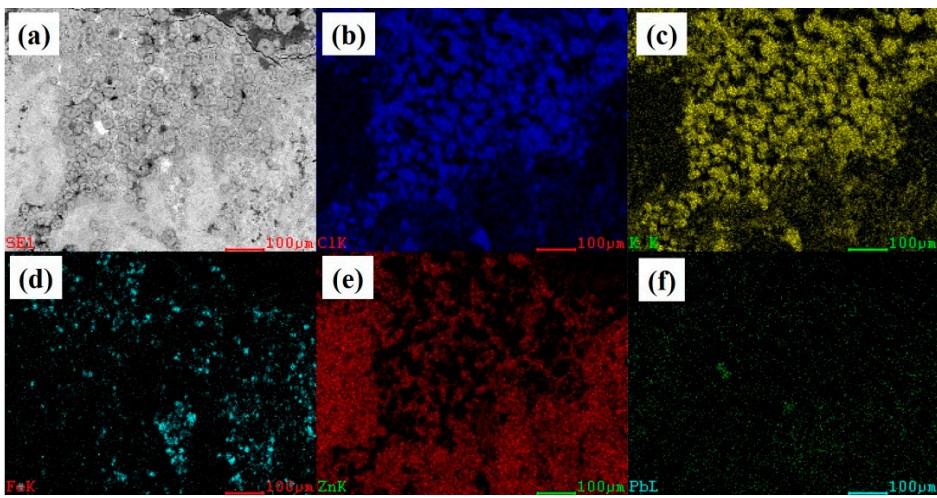

**Figure 4.** Typical morphology of deposition surface.

**Figure 5.** EDS maps of flat and transitional areas. (**a**) EDS map; (**b**) Cl; (**c**) K; (**d**) Fe; (**e**) Zn; (**f**) Pb.

Figure 6 presents the EDS maps results for the porous areas. It was also found that the distribution of Cl (Figure 6b) and K (Figure 6c) overlapped, but no cubic KCl particles appeared. No enrichment of PbO was observed in the porous region. It was regrettable that no $ZnFe_2O_4$ phase was found in any area. From the EDS maps results of the various regions, certain areas existed, in which, the Fe and Zn overlapped, while the $ZnFe_2O_4$ phase was most likely to appear at these positions. The reason for the $ZnFe_2O_4$ phase absence was that the $ZnFe_2O_4$ phase was covered with KCl and ZnO.

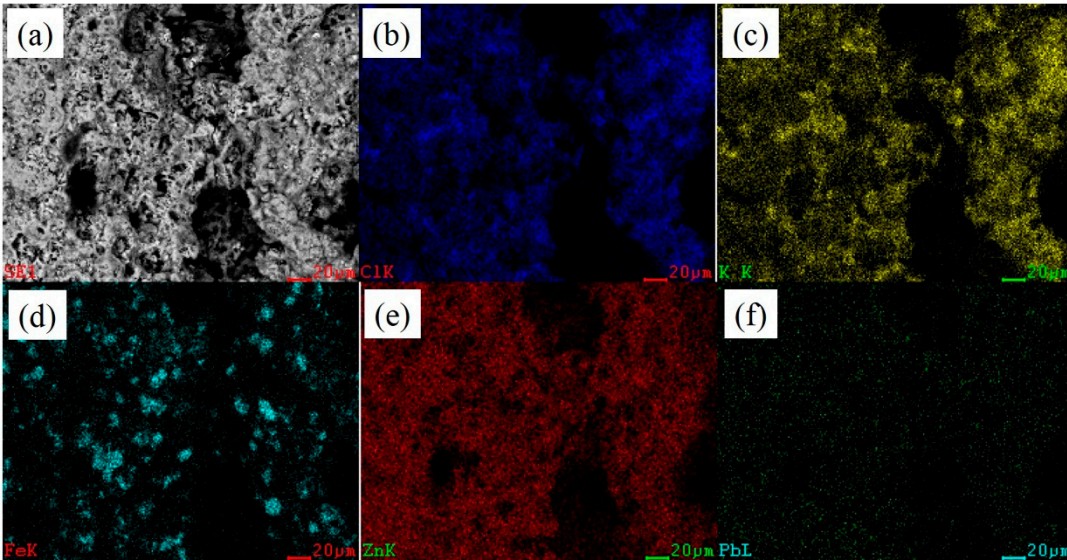

**Figure 6.** EDS maps results for porous areas. (**a**) EDS map; (**b**) Cl; (**c**) K; (**d**) Fe; (**e**) Zn; (**f**) Pb.

Certain filled holes were found on the deposition product surface, as presented in Figure 7a. Following the hole interior enlargement, it was observed that the walls of the hole were covered with molten KCl and cubic KCl particles began to appear (Figure 7b). This was similar to the transitional areas, but these particles were not completely crystallized, compared to the particles of KCl in the flat areas (Figure 7c).

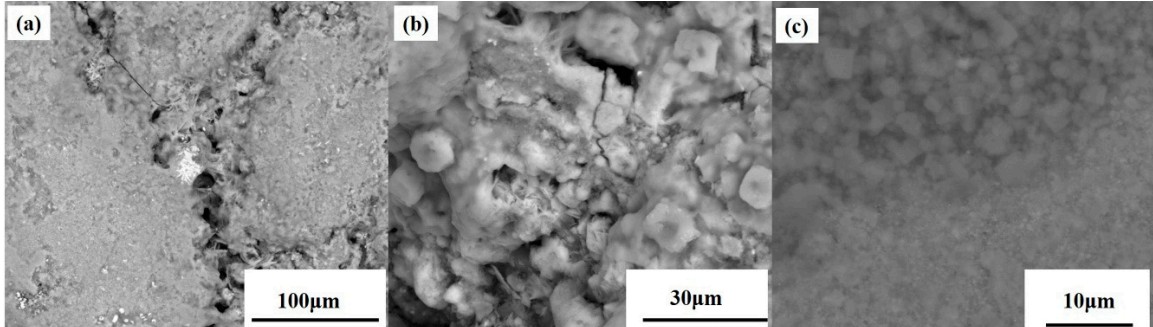

**Figure 7.** Filled holes (**a**), KCl morphology in hole (**b**) and morphology of KCl in flat areas (**c**).

### 3.2. Experimental Results and Discussion

The XRD results of samples of 1–4 groups of experiments in Table 2 are shown in Figure 8. The phases in sample 1 were PbO and ZnO. In addition to the PbO and ZnO phases, $ZnFe_2O_4$ was present in sample 2 due to the addition of $Fe_2O_3$ component to sample 2. Three phases of PbO, ZnO and KCl were present in sample 3, while in sample 4, four phases of PbO, ZnO, KCl and $ZnFe_2O_4$ were present. In the 1–4 groups of experiments, the sample 1 was still powdered following cooling, while the samples in groups 2–4 were bulk after the samples were cooled. Through the XRD results, it could be observed that the mixture of ZnO and PbO did not produce any new compound subsequently to heating, signifying that no cohered phase existed. Therefore, the sample of experiment 1 was still powdery following cooling. The $ZnFe_2O_4$ phase appeared in the sample of experiment 2, while it was possible that the $ZnFe_2O_4$ would bind the other powders together. In experiment 3, although no new compound appeared, the melting point of KCl was low, while the particles of the other substances were adhered to each other during the solidification of KCl. Also, sample 4 presented a result of the combined effect of KCl and $ZnFe_2O_4$. From the 1–4 group of experiments, KCl and $ZnFe_2O_4$ produced the effect of bonding.

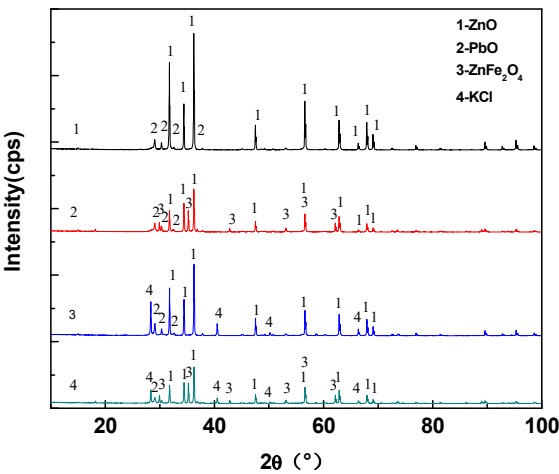

**Figure 8.** Experimental 1–4 sample XRD results.

Figure 9 presents the XRD and SEM results of the two samples in experiment 5, where the water quenched sample was denoted as W and the sample cooled in the air was denoted as A. From the XRD results of the two samples, the $ZnFe_2O_4$ could be completely crystallized under the conditions of rapid cooling in water or slow cooling in the air. This result indicated that the $ZnFe_2O_4$ was solid when it formed and that the melting point of $ZnFe_2O_4$ exceeded 1200 °C. Therefore, the mechanism of $ZnFe_2O_4$ formation reaction was obtained, as presented in Equation (6). Through the FactSage 7.0 thermodynamics software, the $\Delta G_{ZnFe2O4}$ was calculated, as presented in Equation (7):

$$ZnO_{(S)} + Fe_2O_{3(S)} = ZnFe_2O_{4(S)} \tag{6}$$

$$\Delta G_{(ZnFe_2O_4)} = -22723.72902 + 2.90446T \ (873 \ \text{K} - 1523 \ \text{K}) \tag{7}$$

From Equation (7), it could be observed along with Reaction (6) for the endothermic reaction, that the lower temperature was more conducive to the $ZnFe_2O_4$ formation. It can be seen from Figure 3 that there was $Fe_2O_3$ phase in the actual deposition, but there was no $Fe_2O_3$ phase in the sample 5. This may be due to the complete conversion of $Fe_2O_3$ to $ZnFe_2O_4$ during the formation of actual deposition: (1) the contact probability of $Fe_2O_3$ and ZnO particles was lower in the gas phase prior to deposition; (2) Particle of $Fe_2O_3$ and ZnO can be stably contacted after being deposited on the heat exchanger wall, but this will result in decrease of particle temperature. Although $\Delta G_{ZnFe2O4}$ was reduced with decrease of temperature and which was favorable for the formation of $ZnFe_2O_4$. However, with the formation of $ZnFe_2O_4$, the kinetic conditions of the reaction may be destroyed because $ZnFe_2O_4$ will block the contact between $Fe_2O_3$ and ZnO, and resulting in the $Fe_2O_3$ not being fully converted into $ZnFe_2O_4$; (3) the $Fe_2O_3$ on the deposition product originated from the high-sized particles of dust, resulting in the $Fe_2O_3$ and ZnO particles not to be completely in contact. From the SEM-EDS results, many holes existed on the sample surface, but the holes (black) of sample W were dense but low in size, while the holes of sample A appeared sparse but high in size. This might be due to the fact that the W sample rapidly cooled and prevented the contact of different $ZnFe_2O_4$ particles (gray phase in the Figure 9) from increasing in size, while the $ZnFe_2O_4$ retained the initial state at the time of generation. The sample cooling rate was slow for different $ZnFe_2O_4$ crystal contacts and growths to provide the temperature conditions, while the hole size increased due to the $ZnFe_2O_4$ crystal growth. The formation of $ZnFe_2O_4$ and the crystallization of the other powder particles were fixed, for the entire sample to appear massive. This was also the reason for the sample 2 massive appearance.

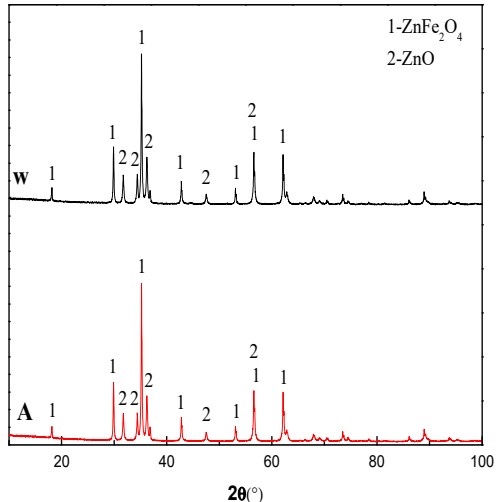 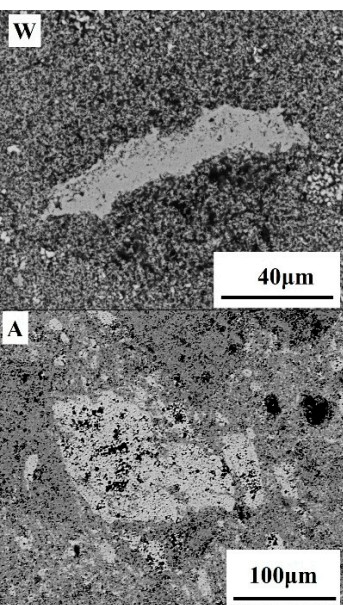

**Figure 9.** Experiment 5 XRD and SEM results of water quenched sample (W) and air cooled sample (A).

The KCl and $ZnFe_2O_4$ bond strength test results are presented in Figure 10. The bond strength of the sample increased as the KCl and $Fe_2O_3$ contents increased. Among them, since the content of KCl was only 5% in sample 6, the sample itself had unbonded powder. Consequently, the bond strength performance was quite low. From the latter, it could be observed that the cohered phase was the $ZnFe_2O_4$, when the $Fe_2O_3$ was added to the ZnO and PbO powder matrix. From the experimental results, the bonding effect of $ZnFe_2O_4$ was significantly improved compared to KCl. When the KCl content was 15%, the bond strength was 17 times/20 cm, while the $ZnFe_2O_4$ was used as the cohered phase. When the $Fe_2O_3$ content reached 10%, the bond strength reached 17 times/20 cm. Figure 11 presents the bonding methods of KCl and $ZnFe_2O_4$, respectively. This was due to different $ZnFe_2O_4$ particle growths to form a macroscopic porous structure, while the other particles were embedded in the hole. Also, in the formation of $ZnFe_2O_4$, a part of the bonded phase was consumed as the ZnO changed into a binder phase. In other words, each micro-$ZnFe_2O_4$ particle adhered to the solid particles in a lower amount compared to KCl bonded particles. Consequently, the performance of $ZnFe_2O_4$ bonding was stronger than KCl. By contrast, in the actual deposition formation, since the formation of $ZnFe_2O_4$ was a solid-solid reaction, the material for the deposition immobilization on the wall of the heat exchanger was KCl. The KCl content in the deposition was approximately 22%, whereas when the KCl content reached 20%, the sample bond strength was 30 times/20 cm, indicating that the KCl existence in the deposition was sufficient to fix the deposition on the wall of the heat exchanger. The KCl bonded other phases through the liquid to solid transition, while the $ZnFe_2O_4$ produced a porous structure to fix the other solid particles. The $ZnFe_2O_4$ was a solid phase that could not self-immobilize along with other substance particles on the heat exchanger walls. Therefore, the KCl was considered as the main cohered phase and the $ZnFe_2O_4$ was the secondary cohered phase.

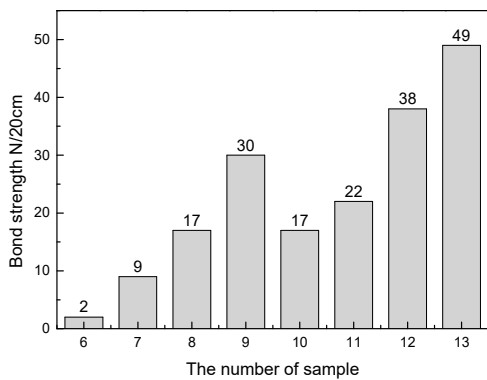

**Figure 10.** Effects of KCl and Fe$_2$O$_3$ content on sample bond strength.

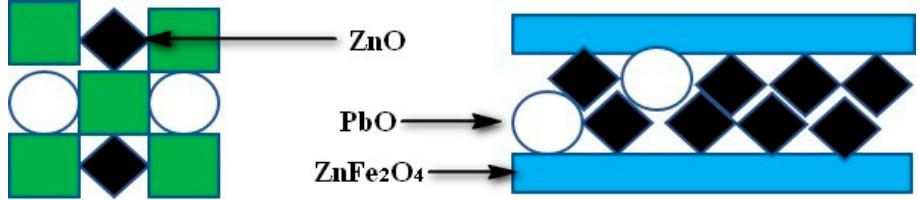

**Figure 11.** Bonding methods of KCl and ZnFe$_2$O$_4$.

### 3.3. Discussion on Formation Mechanism and Prevention of Deposition

Through Chapters 3.1 and 3.2, two types of cohered phases were identified, namely the KCl and ZnFe$_2$O$_4$. The bonded phases were ZnO, PbO and the iron-containing particles. Therefore, it could be inferred that the formation mechanism of the deposition product was: (1) the KCl solidified on the heat exchanger wall, while the iron-containing particles, as well as the ZnO and PbO particles would be bonded with KCl during the KCl solidification. Since the iron-containing particle sizes exceeded the ZnO and PbO sizes, the iron-containing particles would remain on KCl earlier than the ZnO and PbO deposition. (2) the ZnFe$_2$O$_4$ was formed following the contact with ZnO and Fe$_2$O$_3$, while different ZnFe$_2$O$_4$ phases grew to form the porous structure. (4) the KCl in the porous structure continued to solidify and bond other dust particles, for the hole to be filled; (5) When the hole was filled, the formations of gray flat and transition areas occurred, not being completely filled, while retaining the porous area morphology. The process of deposition formation is presented in Figure 12.

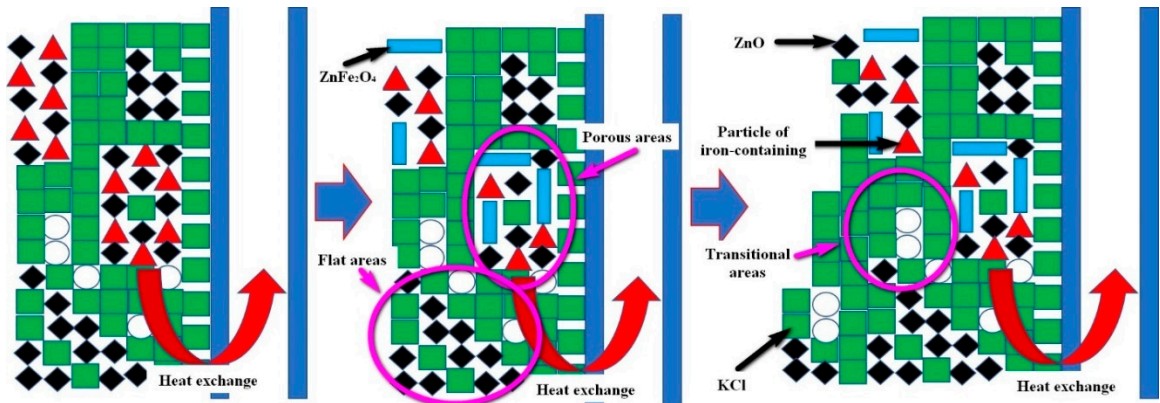

**Figure 12.** Schematic diagram of deposition formation.

In summary, to prevent the formation of deposition product, the appearance of the cohered phase must be avoided. In this study, the cohered phases of the deposition were KCl and ZnFe$_2$O$_4$. In the production of the rotary hearth furnace, the carbonaceous pellets should be prevented from being

dispersed into the flue gas, to cause the iron-containing material and the ZnO to form zinc ferrite. When the rotary hearth furnace was used to treat the metallurgical waste, one of its main functions was to remove and recover both Zn and Pb. Consequently, the presence of ZnO in the flue gas could not be avoided. Therefore, in order to avoid the emergence in the heat transfer system of RHF, it was necessary to avoid the KCl occurrence in the RHF. Guo Zhancheng et al. [16–19] proposed a method of KCl extraction with sintered dust through the sintered dust immersion in water. The KCl dissolved in water during the immersion to form a solution of an insoluble material for the production of iron concentrate, along with a leaching solution for the extraction of KCl. Through the results utilization of Guo Zhancheng et al., a production process to avoid the occurrence of sedimentation was proposed. As presented in Figure 13, the raw material containing KCl was subjected to water immersion treatment prior to pelletizing. Also, the KCl content was maximized within water, producing a leaching solution for the extraction of KCl. Through this process, not only the KCl could be avoided during the production of RHF hazards, but also the maximum use of the waste of the metallurgical process could be achieved.

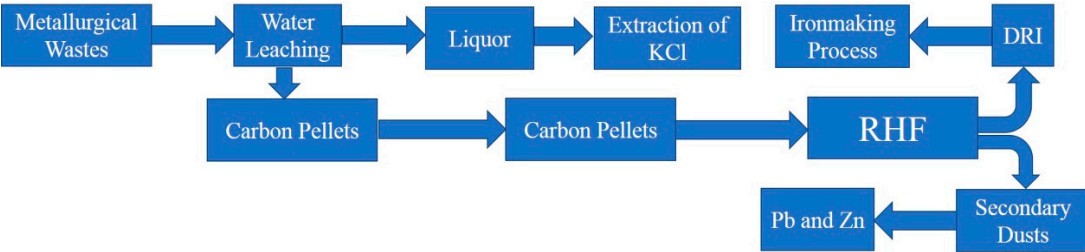

**Figure 13.** A process of RHF without deposition generation.

## 4. Conclusions

Through the detection and experimental study of the deposition in the heat exchanger system, the following conclusions were drawn:

(1) The main cohered phase of the deposition was KCl. The other solid particles were fixed by KCl through the solid-liquid transition of KCl.

(2) The secondary cohered phase of the deposition was $ZnFe_2O_4$, forming a porous structure, while the other solid particles were fixed in the porous structure.

(3) The bond strength of $ZnFe_2O_4$ was better compared to KCl. Under the combined action of KCl and $ZnFe_2O_4$, the deposition could stably occur on the heat exchanger wall.

(4) A new RHF process was proposed to avoid the generation of sediments and to maximize the use of waste from the metallurgical process.

**Author Contributions:** Writing—original draft, Y.P.; Study and design, J.W.; Date interpretation, X.S.; Literature research and date collection; Y.L.

**Funding:** This research was supported by Natural Science Foundation of China (NO. 51874029), The project of State key laboratory of advanced metallurgy (NO. 41618018) and China Postdoctoral Science Foundation (NO. 2018M641195).

**Conflicts of Interest:** The authors declare no conflict of interest.

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
