# Peer review of "Study of the Deposition Formation Mechanism in the Heat Exchanger System of RHF"

_metals, doi:10.3390/met9040443_

Round 1

Reviewer 1 Report

This paper gives really interesting results. It is reasonably well carried out but the SEM-EDX maps are not enough clear. Can they be reproduced with better quality of the photos to improve visibility?

My comment to the Table 2 is that there is no information about the statistical significance of the results shown. The authors give 4 valid digits for each data and that requires some discussion about uncertainties.

The first time you use the abbreviation DRI you need to explain what it means. There is no explanation in the whole paper. Not everyone knows these things.

Figure 8: The figure caption needs to be more informative.

There is a strange sentence in lines 159-160. It starts with "the Fe2O3 phase" and ends with "into ZnFe2O4". The authors should consider rewriting this sentence to make it logical.

The bond strength unit is not known to me. It would be polite to the readers to define it and how it is measured.

The English needs some revision. Not much but just some polishing to make reading smoother.

Author Response

Dear Editors and Reviewers:

Thank you for your letter and for the reviewers’ comments concerning our manuscript entitled “Study of Deposition Formation Mechanism in Heat Exchanger System of RHF” (ID metals-479852). Those comments are all valuable and very helpful for revising and improving our paper, as well as the important guiding significance to our researches. We have studied comments carefully and have made correction which we hope meet with approval. Revised portion are marked in red in the paper. The main corrections in the paper and the responds to the reviewer’s comments are as following:

Responds to the reviewer’s comments:

Reviewer #1:

1.Response to comment:

This paper gives really interesting results. It is reasonably well carried out but the SEM-EDX maps are not enough clear. Can they be reproduced with better quality of the photos to improve visibility?

Response: Considering the Reviewer’s suggestion, we recreated all the images and improved the visibility of the image. We hope that the new image will meet the requirements of editors, reviewers and readers.

2.Response to comment:

My comment to the Table 2 is that there is no information about the statistical significance of the results shown. The authors give 4 valid digits for each data and that requires some discussion about uncertainties

Response: The experimental sample components of Table 2 are designed according to the proportion of various substances in the actual deposition. In order to avoid the confusion of the reader, we added the sentence of explanation on line 63: “components of Table 2 are designed according to the proportion of various substances in the actual deposition

3.Response to comment:

The first time you use the abbreviation DRI you need to explain what it means. There is no explanation in the whole paper. Not everyone knows these things.

Response: Considering the Reviewer’s suggestion, we explained the meaning of DRI on line 34,that is,direct reduction iron is added after the DRI.

4.Response to comment:

Figure 8: The figure caption needs to be more informative

Response: We added the description of Figure 8 in lines 138-142 of the article, and the content as follows: “The XRD results of the experimental samples of 1–4 groups of experiments in Table 3 are shown in Figure 8. The phases in Sample 1 were PbO and ZnO. In addition to the PbO and ZnO phases, ZnFe2O4 was present in sample 2 due to the addition of Fe2O3 component to sample 2. Three phases of PbO, ZnO and KCl were present in sample 3, while in sample 4, four phases of PbO, ZnO, KCl and ZnFe2O4 were present.”

5.Response to comment:

There is a strange sentence in lines 159-160. It starts with "the Fe2O3 phase" and ends with "into ZnFe2O4". The authors should consider rewriting this sentence to make it logical.

Response: We modified the wrong sentence which was pointed out by the reviewer. The modified sentence is as follows “It can be seen from Figure 3 that there was Fe2O3 phase in the actual deposition, but there was no Fe2O3 phase in the sample 5. This may be due to the complete conversion of Fe2O3 to ZnFe2O4 during the formation of actual deposition”.

6.Response to comment:

The bond strength unit is not known to me. It would be polite to the readers to define it and how it is measured.

Response: When testing the strength of the pellet, the pellet falls from a height of 0.5 m, and the falling strength of the pellet is N when the N-th pellet is broken. There is no uniform standard for the bond strength of deposition in previous studies. Therefore, the standard for the falls strength of the pellet is referenced. The standard content is given in lines 76-79: “Subsequently to the sample obtained in experiment 8–13 room temperature cooling, the sample was allowed to fall into a sufficiently large corundum crucible at a height of 20 cm. Following the sample drop, the powder was collected and the bond strength of the sample was N times/20 cm, when the powder exceeded 50% of the total mass subsequently to falling N times.”

7.Response to comment:

The English needs some revision. Not much but just some polishing to make reading smoother

Response: Considering the Editor’s suggestion, this paper has been revised by two native English speakers to ensure the work is reported and discussed clearly.

We tried our best to improve the manuscript and made some changes in the manuscript. We appreciate for Editors/Reviewers’ warm work earnestly, and hope that the correction will meet with approval.

Once again, thank you very much for your comments and suggestions.

Yours sincerely,

Xuefeng She

E-mail: shexuefeng@ustb.edu.cn

       Panyuzhuustb@163.com

Reviewer 2 Report

The studies presented in this work are very interesting. Mertorically, the work is written correctly. However, there are a few comments:

1.       In Figures 5 and 6, the analyzed components are almost invisible

2.       Table 1 (line 80): What does TFe mean in the table?

3.       Reaction 3 is incorrect  - too much iron and oxygen on the left side of the equation

4.       Why KCl is in the products in Figure 13 (at the end of RHF process)  if it has been leached in water before the RHF process

5.       The process is justified, but the leaching of large amounts of dust and the subsequent evaporation of water for the purpose of KCl crystallization will involve a large amount of energy. Is it still worth considering leaching of dust in water and leaching of KCl?

Author Response

Dear Editors and Reviewers:

Thank you for your letter and for the reviewers’ comments concerning our manuscript entitled “Study of Deposition Formation Mechanism in Heat Exchanger System of RHF” (ID metals-479852). Those comments are all valuable and very helpful for revising and improving our paper, as well as the important guiding significance to our researches. We have studied comments carefully and have made correction which we hope meet with approval. Revised portion are marked in red in the paper. The main corrections in the paper and the responds to the reviewer’s comments are as following:

Responds to the reviewer’s comments:

Reviewer #2:

1.Response to comment:

In Figures 5 and 6, the analyzed components are almost invisible

Response: Considering the Reviewer’s suggestion, we recreated all the images and improved the visibility of the image. We hope that the new image will meet the requirements of editors, reviewers and readers.

2.Response to comment:

Table 1 (line 80): What does TFe mean in the table?

Response: In the usual minerals, iron will exist in different oxide forms. In order to measure the utilization value of iron ore, the content of iron is usually expressed by TFe(total iron element content). In actual deposition samples, iron exists in the form of Fe2O3 and ZnFe2O4. The TFe in Table 2 represents all existing forms of iron in the deposition.

3.Response to comment:

Reaction 3 is incorrect - too much iron and oxygen on the left side of the equation

Response: We are sorry for typo of Reaction 3. The new reaction is given as follows:

                         (3)

4.Response to comment:

Why KCl is in the products in Figure 13 (at the end of RHF process)  if it has been leached in water before the RHF process

Response: We are sorry for typo of Figure 13. We removed KCl from the RHF product. The new process chart is shown as follow:

5.Response to comment:

The process is justified, but the leaching of large amounts of dust and the subsequent evaporation of water for the purpose of KCl crystallization will involve a large amount of energy. Is it still worth considering leaching of dust in water and leaching of KCl?

Response: In the raw materials of the RHF process, KCl is mainly present in the sintered dust, and the ratio of the sintered dust as the raw material of the RHF is only 10%-20%. References 16-19 describe the technique of extracting KCl using sintered dust. The process in Figure 3 mainly uses this technique to extract KCl before it is used as a raw material for RHF. The technique described in refs 16-19 consumes energy, but it is still profitable.

We tried our best to improve the manuscript and made some changes in the manuscript. We appreciate for Editors/Reviewers’ warm work earnestly, and hope that the correction will meet with approval.

Once again, thank you very much for your comments and suggestions.

Yours sincerely,

Xuefeng She

E-mail: shexuefeng@ustb.edu.cn

       Panyuzhuustb@163.com
